

# Characteristics of ambulatory care visits to family medicine specialists in Taiwan: a nationwide analysis

An-Min Lynn, Tzu-Chien Shih, Cheng-Hao Hung, Ming-Hwai Lin, Shinn-Jang Hwang and Tzeng-Ji Chen

Department of Family Medicine, Taipei Veterans General Hospital, Taipei, Taiwan
School of Medicine, National Yang-Ming University, Taipei, Taiwan

## ABSTRACT

Although family medicine (FM) is the most commonly practiced specialty among all the medical specialties, its practice patterns have seldom been analyzed. Looking at data from Taiwan's National Health Insurance Research Database, the current study analyzed ambulatory visits to FM specialists nationwide. From a sample dataset that randomly sampled one out of every 500 cases among a total of 309,880,000 visits in 2012, it was found that 18.8% ($n = 116,551$) of the 619,760 visits in the dataset were made to FM specialists. Most of the FM services were performed by male FM physicians. Elderly patients above 80 years of age accounted for only 7.1% of FM visits. The most frequent diagnoses (22.8%) were associated acute upper respiratory infections (including ICD 460, 465 and 466). Anti-histamine agents were prescribed in 25.6% of FM visits. Hypertension, diabetes and dyslipidemia were the causes of 20.7% of the ambulatory visits made to FM specialists of all types, while those conditions accounted for only 10.6% of visits to FM clinics. The study demonstrated the relatively low proportion of chronic diseases that was managed in FM clinics in Taiwan, and our detailed results could contribute to evidence-based discussions on healthcare policymaking and residency training.

Corresponding author
Ming-Hwai Lin,
minghwai@gmail.com

## INTRODUCTION

Family medicine (FM) specialists act as general practitioners and provide primary care services in the community, including solving minor cases, making referrals for major diseases, and providing and promoting preventive health services. Because of an overemphasis on medical specialization over the past several decades, the importance of FM specialists has been increasingly noticed in terms of the benefits they provide to patients (*Ryan et al., 2001*; *Shi et al., 2003*; *Baicker & Chandra, 2004*; *Cooper, 2009a*; *Cooper, 2009b*; *Baicker & Chandra, 2009*). The associated issues, such as the practicing environments of FM specialists and primary care reforms, also affect the government's finances and public health (*Bindman & Majeed, 2003*; *Van Weel & Del Mar, 2004*; *Chou et al., 2007*; *Skinner et al., 2009*; *Chang et al., 2011*; *Katz et al., 2012*). On the other hand, the aging societies and inequalities in health care have increased the complexity of constructing deliberate health policies

(*Beckman & Anell, 2013*; *Jin, 2014*). Comprehensive information about family medicine care is essential to the analysis of existing problems. In the United States, the Primary Care Network Survey and National Ambulatory Medical Care Survey (NAMCS) have been useful in this regard in terms of indicating the number of ambulatory visits to primary care clinics and delineating the national profile of primary care (*Cypress, 1982*; *Cypress, 1983a*; *Cypress, 1983b*; *Cantrell, Young & Martin, 2002*; *Binns et al., 2007*). Information from Canadian physicians has also proved valuable in this regard (*Cunningham et al., 2014*). Other countries have also reported on the working status of practicing FMS (*Okkes et al., 2002*; *Emmanuel, Phua & Cheong, 2004*; *Aboulghate et al., 2013*; *Raza et al., 2014*; *Granja, Ponte & Cavadas, 2014*). Nevertheless, literature looking at the practice of FM on a nationwide basis remains sparse in most countries, including Taiwan.

The purpose of the current study was to explore the nationwide characteristics of ambulatory visits to FM clinics recorded by Taiwan's National Health Insurance (NHI) system in 2012. We analyzed the ages and genders of the patients and physicians, the procedures conducted, diagnoses made, and medications prescribed during these visits. The findings may offer valuable information for future discussions on healthcare policy making and FM residency training programs, and may also provide a foundation for making international comparisons.

## MATERIALS & METHODS

### Database

The NHI program in Taiwan, which started in 1995, provides comprehensive healthcare coverage to more than 99% of the country's residents. The National Health Insurance Administration of the Ministry of Health and Welfare has released all de-identified claims data dating back to 1999 for academic research in the form of the National Health Insurance Research Database (NHIRD; http://w3.nhri.org.tw/nhird/). The conduct of the study had been approved by the institutional review board (IRB) of Taipei Veterans General Hospital, Taipei, Taiwan (2013-04-005E). Because of anonymized data that are publicly available on application, our study is exempt from full IRB review.

### Study population

We performed a descriptive and cross-sectional study by accessing the sampling files for the year 2012 (S_CD20120.DAT and S_OO20120.DAT of NHIRD). In the terminology of NHIRD, the dataset "CD" is defined as the collection of all outpatient visit files, while the "OO" dataset comprises the outpatient order files. According to NHIRD, the size of subset from each month is determined by the ratio of the amount of data to that of the entire year. Then the systemic sampling is performed for each month to randomly choose a representative subset. A sampling database is obtained by combining the subsets from 12 months. The sampling database of S_CD20120 was constructed at first, and then the relative observations in S_OO20120 were drawn out accordingly. These two sampling files, which exclude information on visits to dental clinics and traditional Chinese medicine clinics, contain a total of 619,760 medical records, and were obtained by a 0.2% sampling

**Table 1** Ambulatory visits within Taiwan's National Health Insurance in 2012, stratified by specialty (1/500 sampling).

| Specialty | Number of visits (%) | Cost claimed (%) | Average cost claimed per visit |
|---|---|---|---|
| Family medicine | 116,551 (18.8) | 51,157,561 (8.3) | 439 |
| Internal medicine | 70,615 (11.4) | 42,547,879 (6.9) | 602 |
| Otorhinolaryngology | 67,881 (11.0) | 29,718,395 (4.8) | 438 |
| Pediatrics | 60,717 (9.8) | 31,574,986 (5.1) | 520 |
| Ophthalmology | 37,692 (6.1) | 25,422,291 (4.1) | 674 |
| Obstetrics & Gynecology | 35,697 (5.8) | 19,674,700 (3.2) | 551 |
| All others | 230,607 (37.2) | 418,023,780 (67.6) | 1,813 |
| Total | 619,760 (100.0)[*] | 618,119,592 (100.0) | 997 |

Notes.

[*] The percentage of the ambulatory visits and cost claimed by specialty had been rounded, therefore the % in each of the table didn't give a total of 100% just.

ratio from the CD and OO datasets for 2012. Each individual record included the patient's identification number, birth date, gender, medical facility, date of visit, the specialty of the consulting physician, and up to three diagnosis codes as defined by the International Classification of Diseases, Ninth Revision, Clinical Modification (ICD-9-CM).

From the sampling data, the details of 116,551 ambulatory visits to FM specialists were extracted and analyzed. A list of reimbursable drugs with additional coding in the Anatomical Therapeutic Chemical (ATC) classification system (http://www.whocc.no/atc_ddd_index/) was provided by the National Health Insurance Administration. The basic data of the contracted medical care institutions presented the status of accreditation: academic medical center, metropolitan hospital, local community hospital, or physician clinic. We also analyzed the diagnoses made, procedures conducted, and medications prescribed during the visits to facilities of various levels.

## Statistical analysis

The programming software Perl version 5.20.2 (produced by Perl) was used for data processing, and regular descriptive statistics were displayed.

## RESULTS

Based on the sampling data, of the 619,760 ambulatory visits made in 2012, 18.8% ($n = 116,551$) were made to FM specialists—making FM the most commonly utilized specialty among all physician specialties (Table 1). FM also accounted for 8.3% of insurance claims, with those claims amounting to an estimated NT$309 billion in 2012.

Among the ambulatory visits to FM specialists, 53.1% were made by female patients ($n = 61,974$) and 46.9% were made by male patients ($n = 54,577$). Stratifying the records by age group demonstrated that patients aged 50–59 years had the highest proportion of ambulatory visits to FM specialists among both genders (male: 18.0%, $n = 9,855$; female: 19.2%, $n = 11,955$), followed by patients aged 60–69 years among both genders (male:

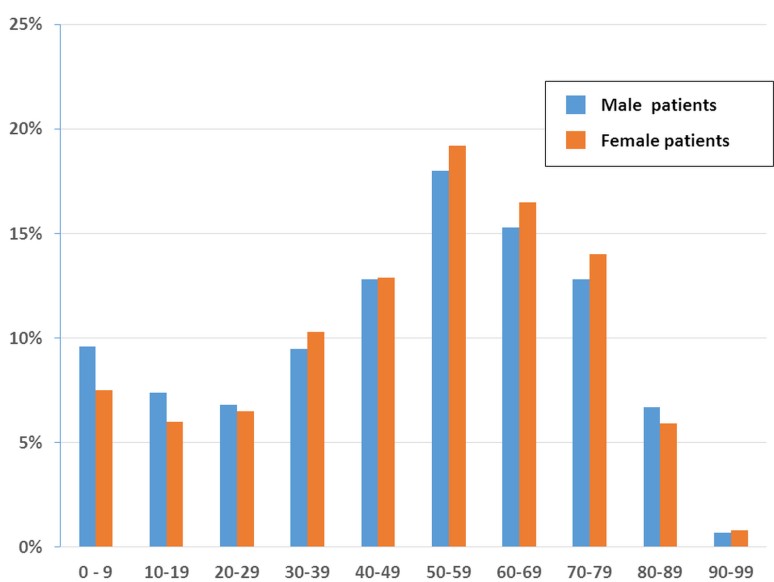

**Figure 1  Age and sex distribution of patients at family medicine specialists within Taiwan's National Health Insurance in 2012 (1/500 sampling).**

15.3%, $n = 8,360$; female: 16.5%, $n = 10,641$) in Fig. 1. No remarkable gap in the number of visits was evident between the various age groups, except that the number of visits by patients aged 80–89 years and 90–99 years dropped obviously from the numbers of visits made by all the other younger groups.

The number of ambulatory visits to FM specialists in terms of the physicians' gender and age is presented in Fig. 2. These data indicated that there are far fewer female physicians than male physicians in all the age ranges. Compared with the other working-age age ranges, the number of female physicians was the highest in the 30–39 years ($n = 3,806$) range, although the number of female physicians in that range was still below the number of male physicians in that age range.

In the current investigation, FM clinics remained the major ambulatory care providers, handling 89.9% ($n = 104,796$) of the ambulatory visits to FM, followed by metropolitan hospitals (4.1%), local community hospitals (3.7%), and academic medical centers (2.3%). Among the ambulatory visits to FM specialists, 65.5% ($n = 76,392$) produced only one diagnosis. The top 10 most common diagnosis groups (based on the first diagnosis code in every medical record) are listed in Table 2. The top diagnosis was acute upper respiratory infection (13.1%), followed by essential hypertension (9.9%), general symptoms (6.9%), acute bronchitis and bronchiolitis (5.9%), and diabetes mellitus (5.7%). The ranking of the diagnosis groups varied according to hospital level.

In general, the most common procedures performed during ambulatory visits to FM were checks of glucose (3.7%, $n = 4,269$), cholesterol, (2.3%, $n = 2,719$), triglyceride (2.2%, $n = 2,599$), S-GPT/ALT (1.9%, $n = 2,270$), serum creatinine (7.8%, $n = 2,802$), and HbA1c (1.5%, $n = 1,781$) levels. The application of the procedures had high consistency between the ambulatory care settings (Table 3).

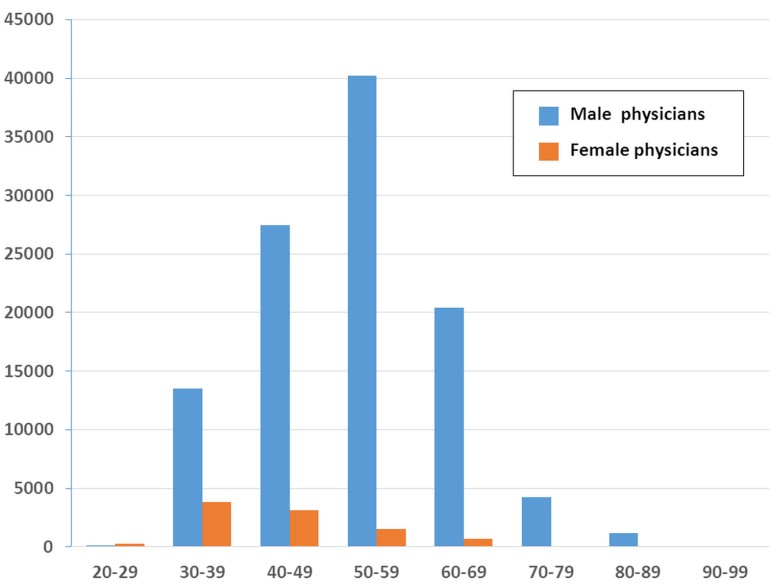

**Figure 2 Age and sex distribution of family medicine specialists within Taiwan's National Health Insurance in 2012 (1/500 sampling).**

**Table 2 Ambulatory visits to family medicine specialists within Taiwan's National Health Insurance in 2012, by disease group and hospital level (1/500 sampling).**

| ICD9CM* | Diagnosis group | Total N = 116,551** | Academic medical center N = 2,637 | Metropolitan hospital N = 4,724 | Local community hospital N = 4,346 | Clinics N = 104,796 |
|---|---|---|---|---|---|---|
| 465 | Acute upper respiratory infections | 22,461 (13.1) | 36 (1.3) | 104 (2.2) | 167 (3.8) | 19,458 (18.5) |
| 401 | Essential hypertension | 11,566 (9.9) | 410 (15.5) | 581 (12.2) | 544 (12.5) | 7,032 (6.7) |
| 780 | General symptoms | 8,097 (6.9) | 127 (3.7) | 81 (1.7) | 141 (3.2) | 4,000 (3.8) |
| 466 | Acute bronchitis & bronchiolitis | 6,909 (5.9) | 15 (0.5) | 46 (0.9) | 99 (2.2) | 5,070 (4.8) |
| 250 | Diabetes mellitus | 6,692 (5.7) | 359 (13.6) | 606 (12.8) | 543 (12.4) | 3,235 (3.0) |
| 272 | Disorders of lipoid metabolism | 5,980 (5.1) | 217 (8.2) | 238 (5.0) | 110 (2.5) | 1,026 (0.9) |
| 460 | Acute nasopharyngitis | 4,440 (3.8) | 56 (1.6) | 19 (0.4) | 40 (0.9) | 3,456 (3.2) |
| 535 | Gastritis & duodenitis | 4,090 (3.5) | 14 (0.5) | 18 (0.3) | 30 (0.7) | 2,285 (2.1) |
| 784 | Symptoms involving head & neck | 3,893 (3.3) | 6 (0.2) | 12 (0.2) | 18 (0.4) | 2,347 (2.2) |
| 729 | Other disorders of soft tissues | 3,778 (3.2) | 7 (0.2) | 9 (0.1) | 33 (0.7) | 2,463 (2.3) |

**Notes.**

* The international classification of diseases, 9th revision, clinical modification.

** Including 48 home care visits.

Of the ambulatory visits to FM specialists, 89.4% ($n = 104,171$) included the prescribing of medication. Approximately 55.8% of the visits where medication was prescribed recorded prescriptions of three or more drugs (one drug 15.8%, two drugs 17.8%, three drugs 19.6%, four drugs 16.9%, five drugs 10.4%, six drugs 5.0%, seven or more drugs 3.9%). The most commonly prescribed medications were anti-histamines for systemic use (25.6%), non-steroid anti-inflammatory and anti-rheumatic products (24.6%), and other analgesics and anti-pyretics (24.5%) (Table 4).

**Table 3** Top ten procedures and laboratory tests prescribed by family medicine specialists within Taiwan's National Health Insurance in 2012 (1/500 sampling).

| NHI code[*] | Procedure | No. of visits | % |
|---|---|---|---|
| 09005C | Glucose | 4,269 | 3.7% |
| 09001C | Cholesterol, total | 2,719 | 2.3% |
| 09004C | Triglyceride | 2,599 | 2.2% |
| 09026C | S-GPT/ALT | 2,270 | 1.9% |
| 09015C | Creatinine | 2,047 | 1.8% |
| 06505C | HbA1c | 1,781 | 1.5% |
| 09044C | LDL-C | 1,586 | 1.4% |
| 48011C | Change dressing-small (<10 cm) | 1,455 | 1.2% |
| 09025C | S-GOT/AST | 1,195 | 1.0% |
| 09043C | HDL-C | 1,189 | 1.0% |

**Notes.**
[*] Taiwan's National Health Insurance code.

**Table 4** Top ten drug classes prescribed by family medicine specialists within Taiwan's National Health Insurance in 2012 (1/500 sampling).

| ATC code[*] | Drug classification | No. of visits | % |
|---|---|---|---|
| R06A | Antihistamines for systemic use | 29,799 | 25.6 |
| M01A | Anti-inflammatory, non-steroids | 28,707 | 24.6 |
| N02B | Other analgesics and antipyretics | 28,516 | 24.5 |
| R05C | Expectorants | 16,263 | 14.0 |
| R05D | Cough suppressants | 15,121 | 13.0 |
| R05F | Cough suppressants & expectorants combinations | 14,943 | 12.8 |
| A02A | Antacids | 14,558 | 12.5 |
| A03A | Drugs for functional bowel disorders | 10,997 | 9.4 |
| R03C | Adrenergics for systemic use | 8,894 | 7.6 |
| R01B | Nasal decongestants for systemic use | 8,886 | 7.6 |

**Notes.**
[*] Anatomical Therapeutic Chemical code.

## DISCUSSION

In this report, FM clinics remained the primary ambulatory care providers, receiving 89.9% ($n = 104,796$) of the ambulatory visits to FM clinics. This finding reveals that primary medical facilities are responsible for most of the FM specialists' workloads. Indeed, all of the top 10 diseases can be managed by FM clinics. In other words, the character of ambulatory visits meets the expectation of primary medical care for minor conditions. However, the number of visits from patients aged over 80 years has markedly decreased. To our knowledge, multiple comorbidity and polypharmacy are prominent in this age group, and both conditions are prevalent in Taiwan (*Lai et al., 1995*; *Chan, Hao & Wu, 2009*; *Lin, Wang & Bai, 2011*). These issues may have contributed to many patients in this age group being too weak to visit FM clinics and being transferred to other departments after being admitted for severe illnesses or comorbidities. In Taiwan, any

transfer to a lower class accommodation is not popular. Therefore, patients aged over 80 years make fewer visits to FM clinics but make more visits to other types of specialists in hospitals instead. This suggests that developing and implementing a referral system policy for home medicine is vital.

We found among patients aged 30–79 years, women paid more visits to FM specialists than men. Some studies of health seeking behaviors revealed that female patients were more interested in health-related information and paid more attention to their own life, including ambulatory visits (*Ek, 2013*; *Galdas, Cheater & Marshall, 2005*; *Oliver et al., 2005*). On the other hand, among patients aged 0–29 years in our study, males had more visits. It was unclear whether younger male patients had more risky behaviors or were more susceptible to fall ill. Besides, parents might decide the seeking health behaviors of children (*Chen et al., 2012*) and boys usually receive more attention than girls.

In addition, the number of visits to FM clinics regarding chronic diseases such as hypertension (ICD 401), diabetes mellitus (250) and dyslipidemia (272) was less than half the number of such visits to other facilities. This suggests that patients prefer to receive regular medical treatment for chronic diseases in hospitals rather than in FM clinics, and that clinic physicians may not pay particular attention to chronic diseases, possibly due to benefit payments, the large number of waiting patients at the clinic, or a lack of the concept of holistic care. In the NHI program, no matter how much time a physician spends with a patient, the physician receives the same reimbursement. Therefore, doctors generally appear not to do much health counseling or education to identify potential patients with chronic diseases (*American Institute in Taiwan*). The government bodies have to innovate the methods of claims to encourage physicians to pay attention holistically (*Michael, 2006*) so that preventive care can be provided. It is also necessary to educate members of the public so that they understand that the diagnosis and treatment of these diseases can be handled by FM clinics. Surely, since the highest number of patients visiting FM clinics consists of those aged 50–59 years, continuing medical education, with appropriate approaches and referrals, also needs to be ensured.

In our study, female physicians accounted for a small amount of visits to FM clinics. The gender disparity increased with age (Fig. 2). The reason might be that fewer women became physicians in the past (*Taiwan Medical Association, 2015*). Another finding was that the majority of visits were made to FM specialists aged 50–59 years. That is, the FM workforce in Taiwan was aging, a phenomenon also observed in obstetrician-gynecologists (*Lynn et al., 2015*).

Resident doctors may have abundant training in academic areas, but they should also have clear insights regarding real-world experiences and, maybe, related lessons. FM specialists should not only manage upper respiratory infections but should also manage geriatric conditions, demonstrate healthy behaviors, including good diets and exercise, and adequately perform health counseling in their future practice. This may encourage medical students to consider all the facts in choosing a career or calling, rather than working exclusively with the common cold.

As for the subject of medication, anti-histamine agents are prescribed most often. Since the NHI program has limited the use of antibiotics for URI or common colds since 2001, antibiotics did not show up among the top 10 most prescribed medications. Symptomatic agents including NSAIDs, antipyretics, expectorants, and cough suppressants were all used for URI. However, antacids are still prescribed often (12.5%), even though the percentage of gastritis diagnoses (3.5%) was not as high as the medication usage. This is because many patients believe they need antacid to protect their stomachs from medication damage (*Chen, Chou & Hwang, 2003*). To decrease the related expenses and potential drug-drug interactions, it is important to educate both the public and physicians that overuse of antacids is not good for the body and that real GI discomfort needs to be diagnosed and treated separately.

The medical care expenses claimed by FM specialists accounted for 8.3% of the total ambulatory costs (Table 1). On average, one ambulatory visit at FM cost less than at most specialties. It might be attributed to fewer laboratory examinations at FM visits. Similar low cost per visit was also observed at visits to otorhinolaryngologists. URI, the most frequent disease seen at both specialties, might play a role (*Liao et al., 2011*).

The resources used in this study, which were compiled by the National Health Insurance Administration, have imposed limitations on our analysis. For instance, the results do not include self-pay procedures or medicines, such as expensive vaccines and cosmetic medicines. However, since the NHI program covers most diseases and requirements for preventive care, the above issues did not have a significant impact on the characteristics of ambulatory visits recorded in this study, although it should also be noted that our figures do not present a complete picture.

## CONCLUSIONS

In Taiwan, FM is the most common utilized specialty of ambulatory care visits and acute URI is the most common diagnosis seen. The decreasing frequency of visits to FM by patients aged above 80 years is notable. The diseases managed differed between hospitals and FM clinics, especially with regard to chronic diseases. In addition, the high proportion of antacids prescribed during visits requires further study.

### Funding

The data used in this study came from the NHIRD provided by the National Health Insurance Administration, Ministry of Health and Welfare, and managed by the National Health Research Institutes in Taiwan. The interpretation and conclusions contained herein do not indicate those of the National Health Insurance Administration, Ministry of Health and Welfare, or the National Health Research Institutes. This study was supported by grants from the National Science Council (NSC 100-2410-H-010-001-MY3) and Taipei Veterans General Hospital (V104E10-001). The funders had no role in study design, data collection and analysis, decision to publish, or preparation of the manuscript.

## Grant Disclosures

The following grant information was disclosed by the authors:

National Health Insurance Administration, Ministry of Health and Welfare.

National Science Council: NSC 100-2410-H-010-001-MY3.

Taipei Veterans General Hospital: V104E10-001.

## Competing Interests

The authors declare that there are no competing interests.

## Author Contributions

- An-Min Lynn conceived and designed the experiments, contributed reagents/materials/analysis tools, wrote the paper, prepared figures and/or tables.
- Tzu-Chien Shih and Cheng-Hao Hung conceived and designed the experiments.
- Ming-Hwai Lin analyzed the data, contributed reagents/materials/analysis tools, reviewed drafts of the paper.
- Shinn-Jang Hwang analyzed the data.
- Tzeng-Ji Chen analyzed the data, reviewed drafts of the paper.

## Data Availability

The following information was supplied regarding the deposition of related data:

Raw data for this work was obtained by application from the National Health Insurance Research Database, Taiwan (http://nhird.nhri.org.tw/en/index.htm) and may not be shared according to the Database's rules governing use. Access to the data used in this study may be obtained by citizens of the Republic of China who fulfill the requirements of conducting research projects.

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
