# Peer review of "Characteristics of ambulatory care visits to family medicine specialists in Taiwan: a nationwide analysis"

_PeerJ, doi:10.7717/peerj.1145_

## Round 0.1 · original submission · Major Revisions

· Academic Editor

Major Revisions

Authors are advised to address comments raised by reviewers including the important concern on ethical approval. It is a descriptive study but yet the authors included a statistical section. The paper can be made more interesting by comparing observations in Taiwan and those from the West. Please correct the references according to the journal guidelines.

Reviewer 1 ·

Basic reporting

The article adhere to the journal policy and the background and introduction section is sufficient. However there are few grammatical and typo errors.
Figure 1 is not cited in the text. If the author did not want to use it in the text suggest to move figure 1 and make appropriate correction.

Experimental design

The research area is within the scope of the journal and the research question is relevant. However, the author did not explained whether the research has gained ethical clearance to be conducted by appropriate body in Taiwan.

Validity of the findings

There is not much statistical analysis involved in this study since it is a descriptive study. However the data is well tabulated and explanatory. The author is advice the make correction to the conclusion section which does not reflect the main findings of the research.

Additional comments

no comments

Annotated reviews are not available for download in order to protect the identity of reviewers who chose to remain anonymous.

Reviewer 2 ·

Basic reporting

General comments :
line 49 and 59 - spacing error
Introduction :
line 45 -51 :the introduction elaboration is not clear and confusing. The author should explain more clearly and used simple words.

Experimental design

This is just a descriptive study but the author did not mention in detail and clearly how the sampling was done but he did mention it in the abstract. Probably the author should mention in the manuscript (method) as well

In the methodology : The author did mention about analyzing the data using Pearson chi square

Validity of the findings

result:
- the author did not mention or state any result using the Pearson chi square test (mention in methodology)
- table - for each table, the title is too long
- table 1- variable others - there is no value

- have to explain why the % in each of the table does not give a total of 100%.

discussion -
- no discussion was noted for - patients gender and age
as well as physician - gender and age
- discussion line 177-183 - a bit confusing. what health examination are they talking about and there is no table to refer.
- since this data is from the national health insurance - any relevant impact of the finding toward the cost - probably have to discuss a bit

Additional comments

This study probably good but its involve a big number and data. There are few corrections that the author has to add and remove.

---

## Round 0.2 · Minor Revisions

· Academic Editor

Minor Revisions

The paper is acceptable pending further minor revisions as suggested by the reviewer.

Reviewer 1 ·

Basic reporting

The author has made all the corrections and the article met all the criteria.

Experimental design

The research is within the scope of the journal and the methodology is satisfactory and fulfilled the ethical conduct.

Validity of the findings

The results and conclusion is appropriate.

Reviewer 2 ·

Basic reporting

The authors have clarified many of the issues that were brought up
previously. The paper become much easy to understand as compared to before.

There are still a few typo errors occur, at line 197 and table 1(last sentence). Probably the author can look and do the correction.

I would suggest to the author, any asterisk or * sentences should be put below the table not in the paragraph.

I found there was no word of figure : for figure 1 and 2 only number 1 and 2 at the figure section. Probably the author can correct that as well.

Experimental design

no comment

Validity of the findings

no comment

Additional comments

The authors have clarified many of the issues that were brought up
previously. The paper become much easy to understand as compared to before.

There are still a few typo errors occur, at line 197 and table 1(last sentence). Probably the author can look and do the correction.

I would suggest to the author, any asterisk or * sentences should be put below the table not in the paragraph.

I found there was no word of figure : for figure 1 and 2 only number 1 and 2 at the figure section. Probably the author can correct that as well.

---

## Round 0.3 · accepted · Accept

· Academic Editor

Accept

Remaining concerns have been addressed. The paper is now acceptable for publication. Congratulations on the work.